# Circadian rhythm shows potential for mRNA efficiency and self-organized division of labor in multinucleate cells

**Leif Zinn-Brooks**[1]*, **Marcus L. Roper**[2]*

**1** Department of Mathematics, Harvey Mudd College, Claremont, California, United States of America,
**2** Department of Mathematics, UCLA, Los Angeles, California, United States of America

\* lzinnbrooks@hmc.edu (LZ-B); mroper@math.ucla.edu (MLR)

**Data Availability Statement:** All relevant data are within the manuscript and its Supporting information files. Matlab scripts and figures are

## Abstract

Multinucleate cells occur in every biosphere and across the kingdoms of life, including in the human body as muscle cells and bone-forming cells. Data from filamentous fungi suggest that, even when bathed in a common cytoplasm, nuclei are capable of autonomous behaviors, including division. How does this potential for autonomy affect the organization of cellular processes between nuclei? Here we analyze a simplified model of circadian rhythm, a form of cellular oscillator, in a mathematical model of the filamentous fungus *Neurospora crassa*. Our results highlight a potential role played by mRNA-protein phase separation to keep mRNAs close to the nuclei from which they originate, while allowing proteins to diffuse freely between nuclei. Our modeling shows that syncytism allows for extreme mRNA efficiency—we demonstrate assembly of a robust oscillator with a transcription rate a thousand-fold less than in comparable uninucleate cells. We also show self-organized division of the labor of mRNA production, with one nucleus in a two-nucleus syncytium producing at least twice as many mRNAs as the other in 30% of cycles. This division can occur spontaneously, but division of labor can also be controlled by regulating the amount of cytoplasmic volume available to each nucleus. Taken together, our results show the intriguing richness and potential for emergent organization among nuclei in multinucleate cells. They also highlight the role of previously studied mechanisms of cellular organization, including nuclear space control and localization of mRNAs through RNA-protein phase separation, in regulating nuclear coordination.

## Author summary

Circadian rhythms are among the most researched cellular processes, but limited work has been done on how these rhythms are coordinated between nuclei in multinucleate cells. In this work, we analyze a mathematical model for circadian oscillations in a multinucleate cell, motivated by frequency mRNA and protein data from the filamentous fungus *Neurospora crassa*. Our results illuminate the importance of mRNA-protein phase separation, in which mRNAs are kept close to the nucleus in which they were transcribed, while proteins can diffuse freely across the cell. We demonstrate that this phase separation

publicly available at: github.com/lzinnbrooks/
circRhythmSyncytia_ZinnRoper21.

**Funding:** The authors acknowledge financial
support from the National Science Foundation
under grant MCB-1840273 (to MR). The funders
had no role in study design, data collection and
analysis, decision to publish, or preparation of the
manuscript.

**Competing interests:** The authors have declared
that no competing interests exist.

allows for a robust oscillator to be assembled with very low mRNA counts. We also investigate how the labor of transcribing mRNAs is divided between nuclei, both when nuclei are evenly spaced across the cell and when they are not. Division of this labor can be regulated by controlling the amount of cytoplasmic volume available to each nucleus. Our results show that there is potential for emergent organization and extreme mRNA efficiency in multinucleate cells.

## Introduction

Syncytia, or multinucleate cells, are present throughout the human body, as muscle cells and bone forming cells, as well as in embryos [1–3]. They also occur in every biosphere and across the kingdoms of life, including fungi, slime molds, and water molds [4–7]. Yet, despite their ubiquity in nature, we do not know how closely cellular processes within syncytia, such as nuclear division, growth, or secretion of enzymes, resemble or diverge from processes in uninucleate cells [8]. In particular, even when bathed in a common cytoplasm, the nuclei of syncytial fungi are capable of dividing autonomously, migrating independently across the syncytium, and expressing different genes [4]. There are many possible ways for a syncytium to divide the labor of making mRNAs among its nuclei. Does coordination of mRNA production require top-down control of nuclei by the organism, or can it emerge spontaneously even when nuclei respond autonomously to cues from the cell's environment?

The genetics of circadian rhythms are among the most throughly dissected of cellular processes [9], and we will use them as a paradigm for how labor of producing mRNAs may be divided between nuclei. The rhythms are fundamental to the life of the cell, regulating timing of the cell cycle and sleep-wake cycle, while also influencing cell physiology, metabolism, and behavior [10–13]. Circadian clocks can be entrained by external cues such as light and temperature, but are also capable of persisting in the absence of these cues [14]. Many circadian rhythms are characterized by biochemical oscillations (such as fluctuations in mRNA and protein concentrations) with period $\sim$ 24 h, and circadian clocks are typically regulated by transcription-translation feedback loops [14–16].

The filamentous fungus *Neurospora crassa* alternates between growth during the day and spore production at night [16, 17]. Circadian timekeeping is regulated by the clock gene *frq* (frequency) and its interactions with WCC (White Collar Complex). Interlocking positive and negative feedback loops drive oscillations of the *frq* gene: in the positive feedback loop, WCC enters the nucleus and activates *frq* transcription. *frq* mRNA is then translated to FRQ protein, which promotes the accumulation of WC-1 and WC-2, the proteins that comprise the White Collar Complex. In the negative feedback loop, FRQ promotes phosphorylation of WCC, which inactivates the complex, thereby preventing it from activating *frq* transcription [13, 18].

Ordinary differential equation models have shown that the known interactions between FRQ and WCC mRNAs and proteins are sufficient to drive *Neurospora's* circadian rhythm. Tseng et al. [18] developed a comprehensive model of the *Neurospora* circadian clock, including every key clock component. The authors showed that their model is capable of reproducing a wide variety of clock characteristics, including a consistent period under constant light conditions, as well as entrainment to photoperiods. They then isolated the crucial components that influence the period and amplitude of oscillations. Dovzhenok et al. [13] formulated a simpler model to study glucose compensation of the *Neurospora* circadian clock, and also to investigate the effect of molecular noise on the robustness of FRQ protein oscillations. Bellman

et al. [19] showed that a *Neurospora* model including only equations for *frq* mRNA and protein, WCC, and WCC:FRQ complex can accurately simulate light-dependent circadian rhythm phase shifts.

These models are capable of producing qualitatively correct time-varying amounts of FRQ and WCC proteins. However, recent data, in which single molecule Fluorescence In-Situ Hybridization (smFISH) was used to map distribution of *frq* mRNAs, indicate that the mRNAs that drive the circadian rhythm are at far lower densities than can support oscillations in the existing ODE models. Remarkably, these data show that at peak transcription, there may be only 6 copies of mRNA per nucleus (Brad Bartholomai, personal communication, 2019). Moreover, low abundance mRNAs are likely to be strongly affected by Poisson noise; do fluctuations in mRNA density affect the precision of the clock?

Prior modeling of Neurospora's circadian rhythm has also omitted the syncytial context, treating the fungal nuclei as a single compartment and the cytoplasm as a second compartment [13, 18–20]. The models are therefore silent on how the elements of the oscillator are assembled across tens, hundreds, or possibly even thousands of nuclei. Indeed, competing hypotheses can be advanced: averaging mRNA and protein abundances in many nuclei may reduce the effect of fluctuations, or, if different oscillators interfere, it may lead to less tightly controlled oscillations.

Although models directly addressing *Neurospora's* circadian rhythm qualitatively capture its clock components, there are too many unmeasured parameters in these models to use them to make quantitative predictions. For this reason, we analyze the syncytial version of a simpler rhythm, which uses a single negative feedback loop to time its clock. In using this model, we limit ourselves to making only semi-quantitative predictions about the real clock. Nonetheless, our model resolves the interplay of Poisson noise, the need to synchronize multiple nuclei, and differential sharing of mRNAs and proteins between nuclei based on their different mobilities within the cytoplasm.

Our syncytial cell model is adapted from Wang and Peskin's [21] single cell model for the mammalian circadian rhythm, which is based on the abundance of PER (PERIOD) proteins. In this model, a single negative feedback loop maintains the clock, driving circadian oscillation of *Per* mRNA and protein levels. The circadian oscillation manifests as a limit cycle, which is attained only above a critical rate of mRNA transcription. Wang and Peskin [21] considered the destabilizing effect of Poisson noise on this limit cycle, as well as showing how the model can be modified to incorporate entrainment by light. In this work, we ask whether stable oscillations can be achieved with lower mRNA costs in a syncytial organism. Along the way, our model signals the existence of a potential general benefit to syncytial organization by allowing predictable protein abundances to emerge from mRNAs with low and fluctuation-affected transcription rates because of the pooling of proteins between nuclei. Mathematically, we must go beyond existing models, which incorporate only temporally varying protein and mRNA concentrations, to model the distribution of mRNAs and proteins within the cell. Our model operates in a regime dominated by the effects of Poisson noise, with mRNA copy numbers, matched to experiments, one or two orders of magnitude smaller than those in previous models.

We begin by describing our mathematical model for circadian rhythms in a syncytial cell. We then run stochastic simulations of our model (using the Gillespie algorithm [22]) for a single nuclear compartment with transcription rates several orders of magnitude below the parameter value used by Wang and Peskin [21], matching the mRNA abundances seen in real fungal cells. We show that stochastic transcription can maintain quasi-periodic limit cycles for transcription rates far below the deterministic threshold at which Wang and Peskin [21] first see limit cycles emerging. We develop a quantitative measure for evaluating the quality of

model circadian limit cycles, and measure this "quality factor" for a wide range of transcription rates. Subsequently, we turn to a syncytial context, and show that protein diffusion between nuclear compartments regulates circadian rhythms in a syncytium by demonstrating that limit cycles are more "organized" (i.e., have a higher quality factor) in a model syncytium than in a uninucleate cell with the same mRNA and protein expression levels. Finally, we demonstrate that protein diffusion also has an "entrainment" effect on our model syncytial cell by comparing protein oscillations in linked nuclear compartments to oscillations in independent nuclei.

## Mathematical model

How is the circadian clock coordinated between nuclei in a multinucleate cell such as *Neurospora*? To address this question, we adapt for syncytial cells the single negative feedback model formulated by Wang and Peskin [21], which was originally proposed for the mammalian circadian oscillator. Our model represents a simplified form of the timing machinery in real *Neurospora* cells since it incorporates only a negative feedback loop, so our predictions will be semi-quantitative at best. Nonetheless, our model allows us to incorporate the elements that are most important for this analysis: stochastic fluctuations and very small mRNA copy numbers.

Our model syncytium is a line of length $L$, divided into $N$ compartments of equal length (later, we will consider non-uniform compartment lengths). Each compartment (index $i = 1$, $2, \ldots, N$) consists of a nucleus ($n$) at its center and a surrounding cytoplasm ($c$), with volumes $V_n$ and $V_c$, respectively. There are four state variables, mRNA $M$ and protein $P$, each present in the nucleus and cytoplasm. Only proteins within the boundaries of a compartment (i.e., in the local cytoplasm) can be imported into that compartment's nucleus. Our notation is

$$M_n^{(i)} = \text{concentration of nuclear mRNA in compartment } i,$$

$$M_c^{(i)} = \text{concentration of cytoplasmic mRNA in compartment } i,$$

$$P_c^{(i)} = \text{concentration of cytoplasmic protein in compartment } i,$$

$$P_n^{(i)} = \text{concentration of nuclear protein in compartment } i.$$

Nuclear mRNAs are transcribed at maximum rate $\alpha$; this transcription is inhibited by nuclear protein. Nuclear mRNAs are also exported out of the nucleus and into their local cytoplasm (at rate $\gamma_m$), where they translate protein (at rate $\beta$) and decay (at rate $\delta_m$). Diffusion of cytoplasmic mRNAs is limited due to their relatively large molecular size [23] and is potentially further reduced by specific interactions between mRNAs and proteins that confine mRNAs within high viscosity RNP droplets that limit their mobilities within the cell [24]. Hence, we assume that cytoplasmic mRNAs remain in the compartment containing the nucleus from which they originated. Within each compartment, our ODEs for $M_n$ and $M_c$ are the same as in [21]:

$$\frac{dM_n^{(i)}}{dt} = \underbrace{\frac{\alpha}{V_n}\left(\frac{K}{K + P_n^{(i)}}\right)^r}_{\text{transcription}} - \underbrace{\gamma_m M_n^{(i)}}_{\text{export}}, \tag{1}$$

$$\frac{dM_c^{(i)}}{dt} = \underbrace{\gamma_m \left(\frac{V_n}{V_c}\right) M_n^{(i)}}_{\text{export}} - \underbrace{\delta_m M_c^{(i)}}_{\text{decay}}. \tag{2}$$

For the derivation of the term for nuclear mRNA transcription, see [21].

Cytoplasmic proteins are imported into their local nucleus (at rate $\gamma_p$), where they decay (at rate $\delta_p$). The ODE for cytoplasmic protein in each compartment is

$$\frac{dP_c^{(i)}}{dt} = \underbrace{\beta M_c^{(i)}}_{\text{translation}} - \underbrace{\gamma_p P_c^{(i)}}_{\text{import}}.$$

(3)

In contrast to cytoplasmic mRNAs, cytoplasmic proteins can also move between compartments via diffusion. We make the simplifying assumption that diffusion of proteins is "fast" relative to the rates of protein translation and import (for details on this assumption and its implications for our model, see the analysis in S1 Appendix). Hence, following either one of these reactions, the distribution of cytoplasmic proteins in our syncytium instantaneously reaches equilibrium (i.e., a uniform concentration of proteins across the entire cell). This is achieved by averaging the concentration of cytoplasmic proteins across all compartments at each time step of numerical simulations, and then assigning this average concentration to each compartment:

$$P_c^{(i)} := \frac{1}{N} \sum_{i=1}^{N} P_c^{(i)}.$$

(4)

In our stochastic simulations, we impose that whenever the total number of proteins in the cytoplasm $p_{\text{tot}}$ changes (via translation or import into the nucleus), the proteins are redistributed so that each compartment contains $\lfloor p_{\text{tot}}/N \rfloor$ proteins, with the remaining proteins (if any) randomly assigned to separate compartments. We use the same equation as [21] for protein concentrations in each nucleus:

$$\frac{dP_n^{(i)}}{dt} = \underbrace{\gamma_p \left( \frac{V_c}{V_n} \right) P_c^{(i)}}_{\text{import}} - \underbrace{\delta_p P_n^{(i)}}_{\text{decay}}.$$

(5)

Most of our analysis in the next section will discuss mRNA and protein counts rather than concentrations—we use $m$ and $p$, respectively, to denote these counts. See Fig 1 for a schematic of our model.

## Results

### Stochastic transcription maintains limit cycles below the threshold for deterministic oscillations

Wang and Peskin [21] showed that their deterministic model produces sustained (rather than damped) oscillations only above a critical peak rate of transcription:

$$\alpha > 4 \cdot 5^5 \cdot v^2 \cdot \frac{KV_n}{\beta},$$

(6)

where, for simplicity, $\gamma_m$, $\gamma_p$, $\delta_m$, and $\delta_p$ are all set equal to $v$. With the default parameter values used by [21] ($v = 2\pi/22 \ h^{-1}$, $V_n = 0.1$ pL, $V_c = 2$ pL, $\beta = 10 \ h^{-1}$, $K = 200$/pL, $r = 5$), (6) predicts that a limit cycle will be maintained for $\alpha > 2039 \ h^{-1}$. While the authors ran many stochastic simulations using their default transcription rate ($\alpha = 180000 \ h^{-1}$), they did not explore the behavior of the stochastic model when $\alpha$ is below the deterministic threshold for a limit cycle. Using the Gillespie algorithm [22], we ran trials of the Wang and Peskin [21] model (i.e., our mathematical model with $N = 1$ compartment) for values of $\alpha$ three and four orders of magnitude below the default rate ($\alpha = 180 \ h^{-1}$ and $\alpha = 18 \ h^{-1}$, respectively). Both of these rates are in

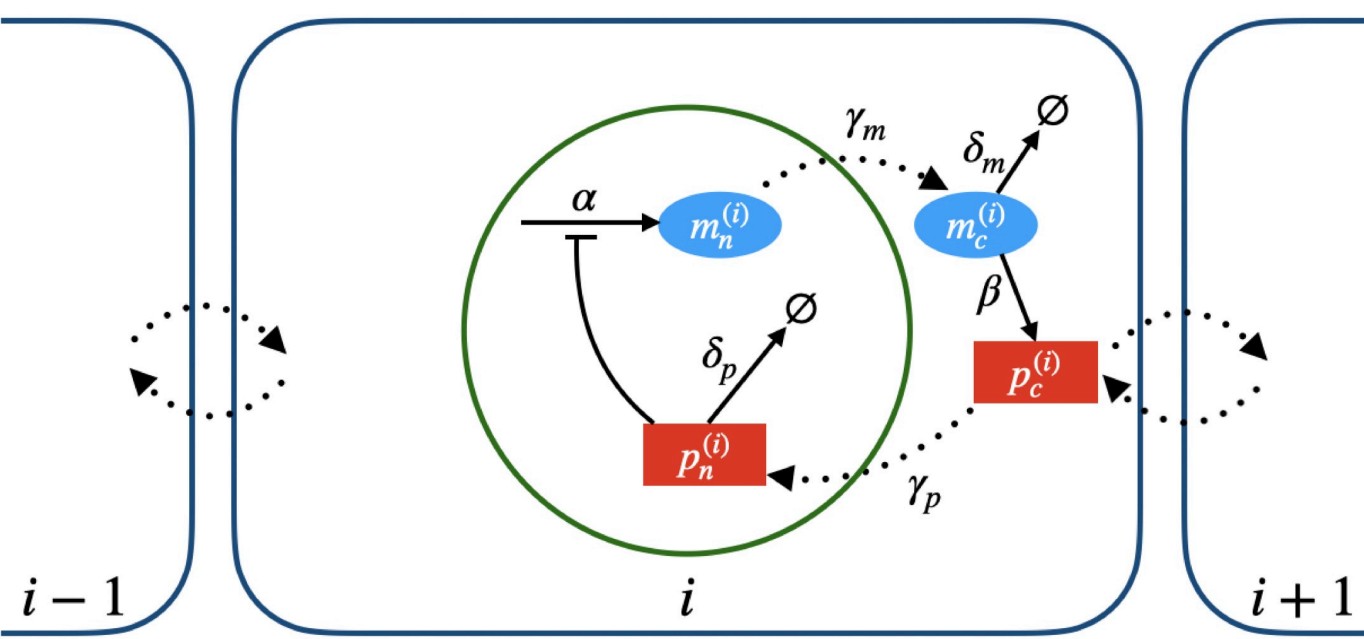

**Fig 1. Schematic of the syncytial circadian rhythm model.** mRNA is indicated by blue ovals and protein by red rectangles. Flows to the empty set $\emptyset$ indicate decay and dotted arrows indicate movement, including import, export, and diffusion. Only proteins move between compartments. Dynamics are the same within each compartment—a single compartment ($i$) is illustrated here.

a regime where the deterministic model displays damped oscillations. Interestingly, while oscillations of protein and mRNA levels in the deterministic model rapidly decay, oscillations are maintained indefinitely in the stochastic model (Fig 2).

In Fig 3, we display time course data from a single stochastic simulation for $\alpha = 18\ h^{-1}$, the lowest transcription rate we tested, and for $\alpha = 180000\ h^{-1}$, the default parameter value used by Wang and Peskin [21]. On average, we find that mRNA counts are about an order of magnitude higher for $\alpha = 180000\ h^{-1}$, and that counts for $\alpha = 18\ h^{-1}$ are much closer to

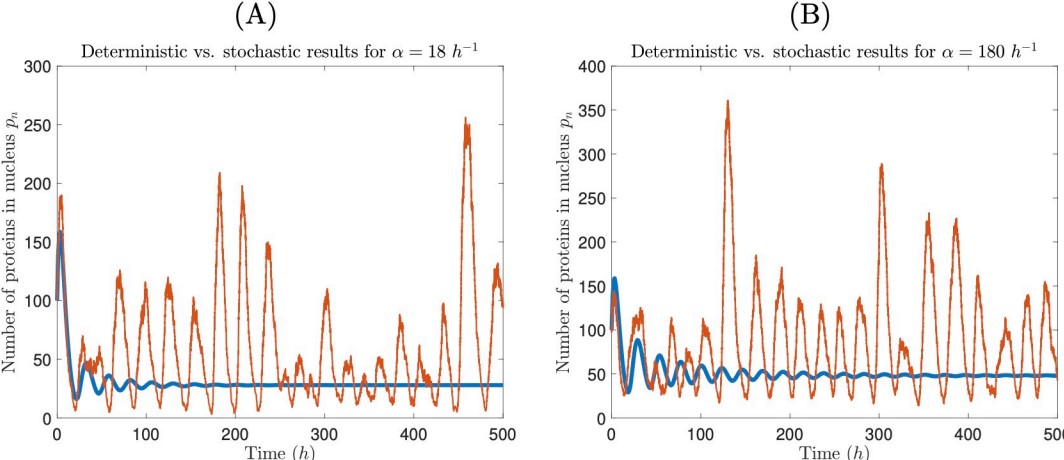

**Fig 2. Stochastic transcription preserves limit cycles.** Deterministic (thick blue curve) vs. stochastic (thin red curve) time course for nuclear protein for $\alpha = 18\ h^{-1}$ (A) and $\alpha = 180\ h^{-1}$ (B). In (A) and (B), $\alpha$ values are two and one orders of magnitude, respectively, below the critical transcription rate derived from (6), so under the deterministic model, oscillations are damped. However, stochastic realizations of the same model support sustained, albeit noisy, oscillations.

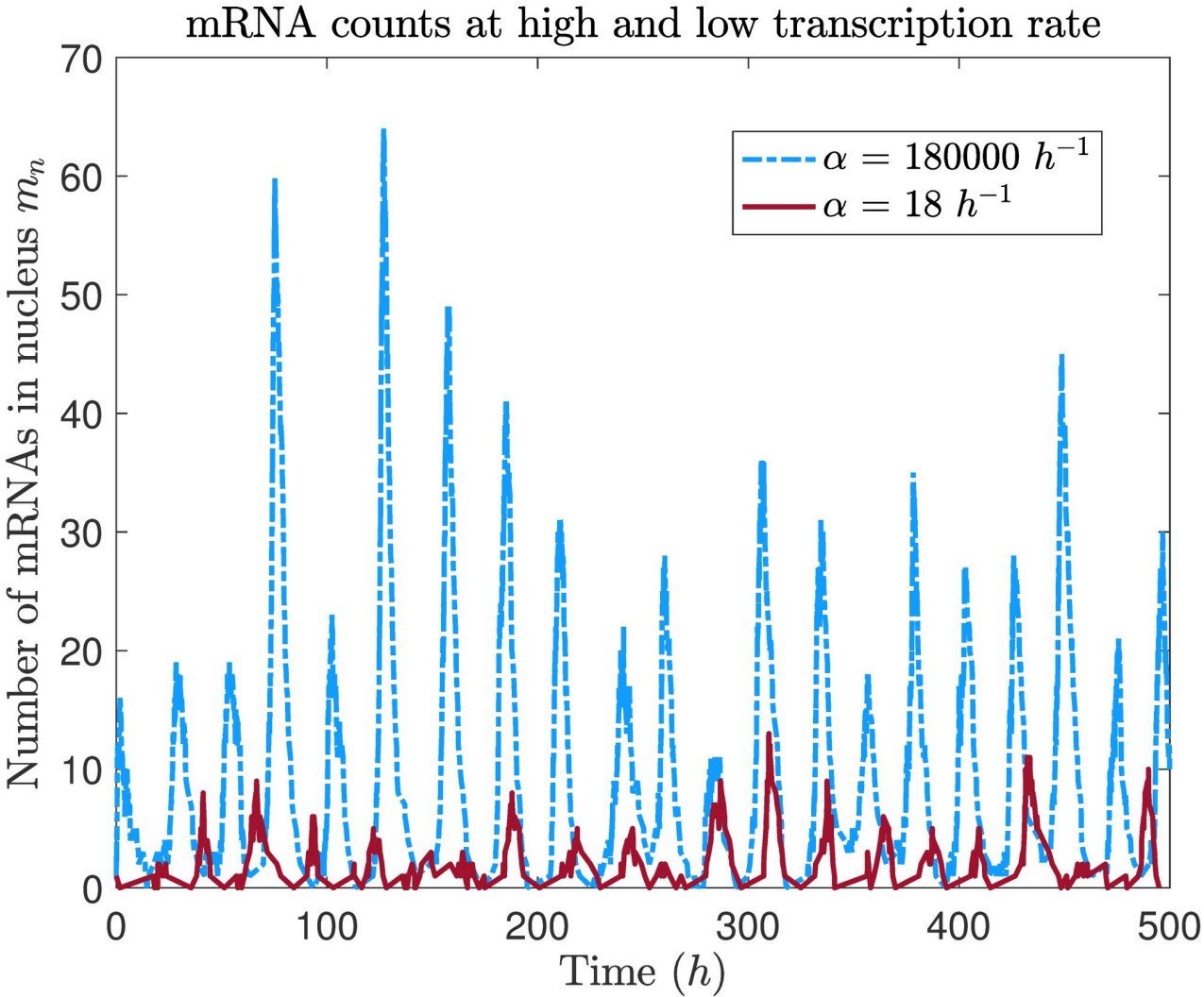

**Fig 3. mRNA oscillations are still evident even at a very low transcription rate.** Time course of number of mRNA copies in the nucleus from individual stochastic simulations using two extremal values of the transcription rate $\alpha$.

experimental data from *Neurospora*. Remarkably, periodic oscillations are still evident in the $\alpha = 18 \ h^{-1}$ case, even with peak mRNA counts in the single digits.

### Measuring the quality of limit cycles

To compare different conditions, we develop a quantitative measure of the "quality" of limit cycles in the stochastic circadian rhythm model. To begin, we find the power spectrum for nuclear protein count $p_n$ over 1000 $h$ of simulated time, averaged over $N = 100$ trials. The power spectrum of a time series gives the power of each frequency component in the signal, computed using Fourier analysis [25]. The peak in the power spectrum indicates the dominant frequency of the signal; a noisy signal that does not have a limit cycle will have a peak of zero. A reliable circadian oscillator should have a consistent period; accordingly, we define the "quality factor" $q$ of our oscillator to be the proportion of the total power that is within a

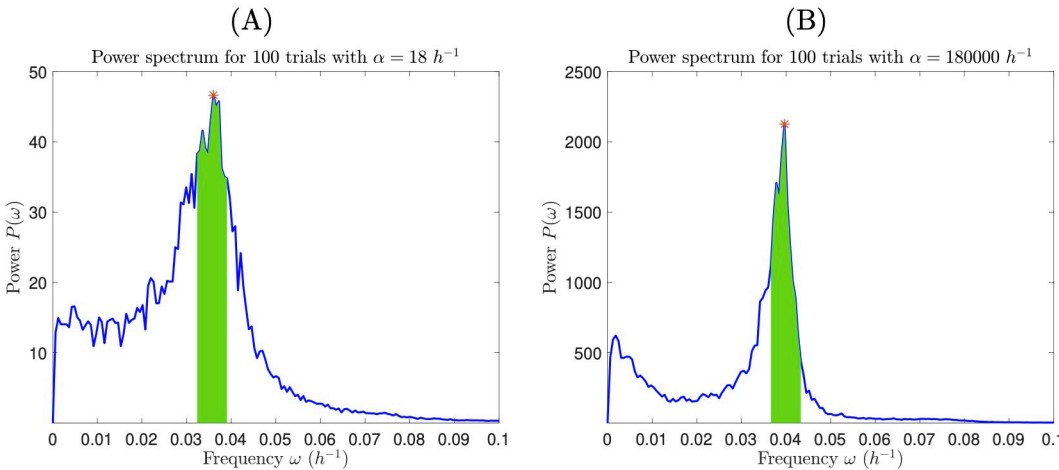

**Fig 4. Power spectrum can be used to measure the periodicity of the circadian rhythm.** We average the power spectrum for nuclear protein over 100 trials, with a low transcription rate (A) and a high transcription rate (B). The quality factor is the fraction of the power spectrum in the interval $[\omega_-, \omega_+]$ (shaded areas) to the total area under the curve. Red asterisks indicate the peak frequency. Quality factors are $q = 0.192$ in (A) and $q = 0.370$ in (B).

certain time $\tau$ of the peak period $T^*$, i.e.,

$$q = \frac{\displaystyle\int_{\omega_-}^{\omega_+} P(\omega)d\omega}{\displaystyle\int_0^{\infty} P(\omega)d\omega}, \tag{7}$$

where $P$ = power, $\omega$ = frequency, and

$$\omega_- = \frac{1}{T^* + \tau}, \quad \omega_+ = \frac{1}{T^* - \tau}. \tag{8}$$

For all subsequent measures of quality factor, we use $\tau = 2\ h$. In Fig 4, we show examples of evaluating quality factor for $\alpha = 18\ h^{-1}$ and $\alpha = 180000\ h^{-1}$.

We compute the quality factor for five orders of magnitude of the transcription rate $\alpha$, ranging from $\alpha = 18\ h^{-1}$ to $\alpha = 180000\ h^{-1}$. We find that quality factor increases with $\alpha$ (see Table 1) since mRNA and protein counts are generally higher for larger values of $\alpha$, making oscillations of nuclear protein less susceptible to Poisson noise. So, although we predict that circadian oscillations can be maintained even when transcription rate is low, oscillations will be more regular in amplitude and period when transcription rate is high. However, gains are modest: quality factor $q$ increases only by a factor of 2 for a $10^4$-fold increase in $\alpha$. We also observe from Fig 4 that changing $\alpha$ has a small, but discernible, effect on the dominant

**Table 1. Maximum transcription rate $\alpha$ vs. quality factor $q$ of nuclear protein oscillations.**

| $\alpha\ (h^{-1})$ | Quality factor ($q$) |
|---|---|
| 18 | 0.192 |
| 180 | 0.227 |
| 1800 | 0.251 |
| 18000 | 0.304 |
| 180000 | 0.370 |

frequency $\omega^*$ of nuclear protein oscillations. $\omega^*$ increases from 0.0360 $h^{-1}$ ($T^* = 27.8\ h$) for $\alpha = 18\ h^{-1}$ to 0.0397 $h^{-1}$ ($T^* = 25.2\ h$) for $\alpha = 180000\ h^{-1}$, and generally increases as $\alpha$ increases.

### Limit cycle quality in a model syncytium

**Uniform compartment sizes.** We now evaluate the quality factor of nuclear protein oscillations for multiple nuclear compartments ($N = 2, 4, 8,$ and 16 compartments) of equal length (i.e., each nucleus contains the same volume of surrounding cytoplasm). As before, we find the power spectrum for 1000 $h$ of simulated time, averaged over 100 trials. Since we expect quality factor to be consistent across compartments, and we would like to compare with the one compartment case, we evaluate the quality factor for a single compartment in each case, rather than for the entire cell. We find that quality factor increases with number of compartments in the syncytium (Fig 5). This is likely because the redistribution of cytoplasmic proteins has an averaging effect on the model, removing some of the noise. As the number of compartments increases, this averaging effect becomes more robust. In fact, the quality factor with 8 or 16

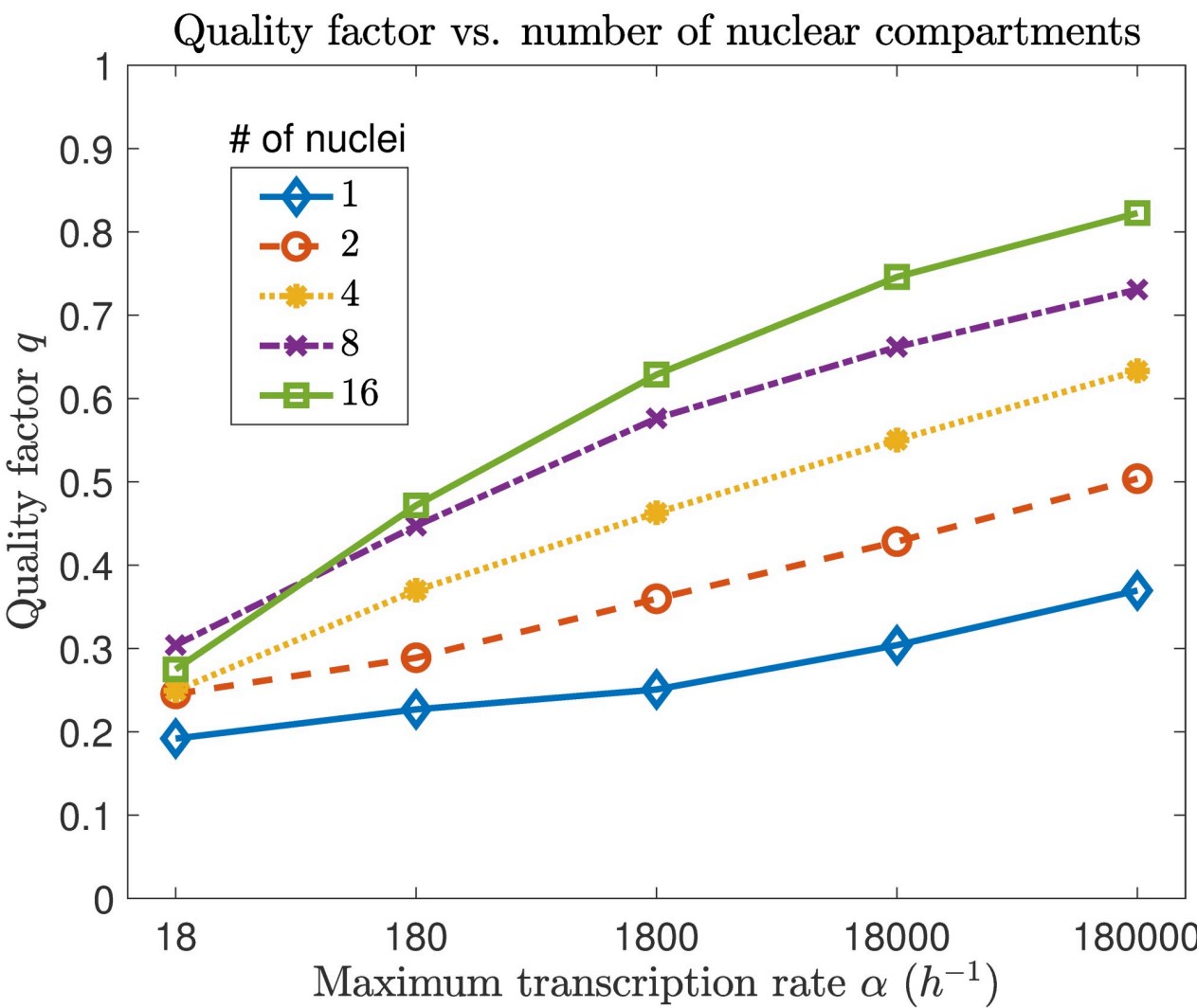

**Fig 5. Quality factor increases with transcription rate and number of compartments.** Quality factor for 1, 2, 4, 8, and 16 nuclear compartments, over five orders of magnitude of the maximum transcription rate $\alpha$.

**Table 2. Peak frequency $\omega^*$ for nuclear protein oscillations (obtained from the power spectrum), with maximum transcription rate $\alpha = 18\ h^{-1}$, for various numbers of nuclear compartments.**

| Number of nuclei | Peak frequency $\omega^*$ $(h^{-1})$ | Peak period $T^*$ $(h)$ |
|---|---|---|
| 1 | 0.0360 | 27.8 |
| 2 | 0.0372 | 26.9 |
| 4 | 0.0391 | 25.6 |
| 8 | 0.0409 | 24.5 |
| 16 | 0.0415 | 24.1 |

nuclear compartments and $\alpha = 18\ h^{-1}$ is comparable to the quality factor for a single compartment with a transcription rate 1000 times higher. We also find that the dominant frequency $\omega^*$ increases with number of compartments (Table 2). This agrees with mathematical modeling results from [26] and [27] (confirmed experimentally in [28]), which showed that frequency increases with coupling strength in a Hill-type repression model for circadian rhythm (the same type of model we study here). In the context of our model, number of nuclear compartments serves as a proxy for coupling strength.

Sharing transcription-inhibiting proteins evenly between nuclear compartments of uniform size results in roughly equal average transcription rates in each nucleus, leading to an improvement of limit cycle quality. But how is the labor of transcribing mRNAs divided between nuclei over each circadian day? To investigate, we ran a stochastic simulation of our model for a model syncytium with two compartments of equal length and with the minimal transcription rate $\alpha = 18\ h^{-1}$, and counted the number of mRNAs transcribed in each compartment over each circadian period. (We define a circadian period as the time interval between troughs in nuclear protein abundances.) The simulation was run for 10000 $h$ (approximately 400 periods). We find that the labor of producing mRNAs often skews strongly toward a single compartment over individual periods (Fig 6). In fact, in nearly 30% of periods, one nucleus produces more than twice the number of mRNAs as the other.

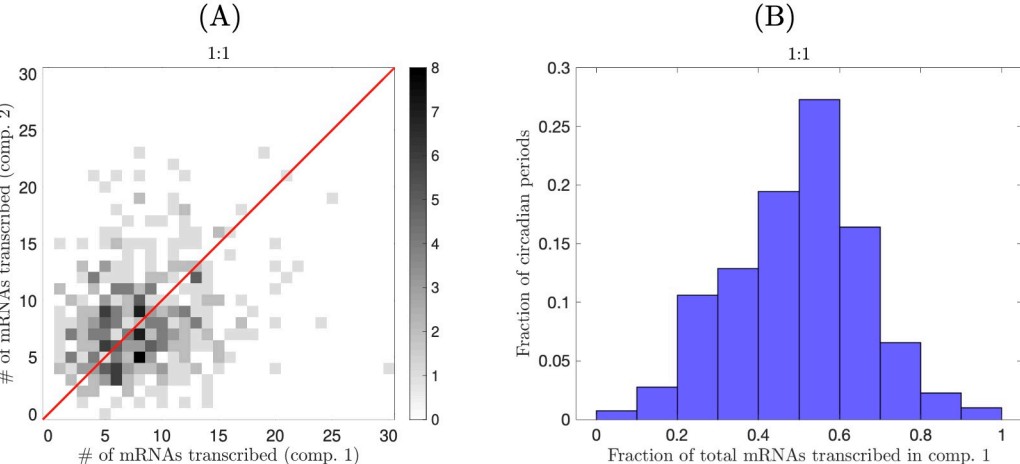

**Fig 6. mRNA production in a model syncytium with uniform compartments.** (A) Number of mRNAs transcribed in each nuclear compartment for approximately 400 circadian periods for a two-nucleus syncytium with uniform compartment sizes (1:1 size ratio). The red line represents equal numbers of mRNAs being produced by each nucleus. (B) Histogram of the results from (A) indicates that one compartment often dominates mRNA production over a single period.

Our model shows that asymmetries in mRNA production can emerge spontaneously from the circadian dynamics. Controlling division of labor may confer selective benefits upon fungi: if one nucleus makes the majority of circadian mRNAs, the other nuclei may be able to devote more time to expressing other mRNAs, potentially boosting overall mRNA outputs. The feedback strength depends on the number of proteins a nucleus encounters, and thus depends on the volume of cytoplasm in that nucleus' compartment. What happens to circadian cycles when these cytoplasmic volumes are not equal (i.e., nuclear compartments are not of uniform size)? We address this question in the next section.

**Non-uniform compartment sizes.** In general, nuclear compartment sizes are not exactly uniform in a syncytial cell (though in some syncytial cells, cytoskeletal elements closely regulate internuclear spacing [29]). In *Neurospora*, nuclear movement and rearrangement constantly modify compartment sizes [30]. For simplicity, we assume that compartment sizes remain constant over time: our model is designed to identify trends in how labor is divided between nuclei, rather than to quantitatively model the real *Neurospora* circadian clock. Because proteins are uniformly distributed in the cytoplasm in our model, the expected number of cytoplasmic proteins in compartment $i$ at time $t$ in our stochastic model is

$$E[p_c^{(i)}(t)] = \frac{l_i}{\sum_{i=1}^{N} l_i} p_{\text{tot}}(t), \tag{9}$$

where $l_i$ is the length of compartment $i$. By examining Eqs (1), (5) and (9), we can infer the effect of compartment size on mRNA transcription. Since larger compartments generally contain more cytoplasmic proteins than smaller ones, it follows that more proteins are imported into nuclei contained within larger compartments. Thus, transcription rates are inhibited more in larger compartments, meaning that mRNA levels should decrease as compartment size increases. To verify this hypothesis, we ran simulations of a two compartment syncytium in which the larger compartment was double the length of the smaller. We find that the smaller compartment contains dramatically more mRNAs than the larger, in both the nucleus and surrounding cytoplasm (Fig 7). Hence, labor of transcription is unevenly divided between compartments, and nuclei in small compartments carry more of the burden of producing mRNAs.

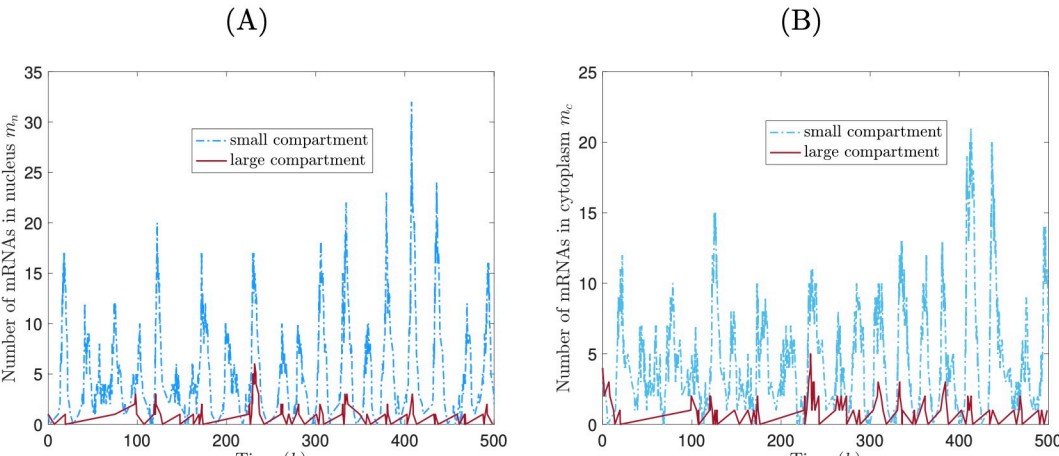

**Fig 7. Nuclei in smaller compartments do more labor of producing mRNAs.** mRNA time courses for nucleus (A) and cytoplasm (B) from a stochastic simulation of a two-nucleus syncytium in which one compartment is twice the length of the other, for $\alpha = 180 \ h^{-1}$.

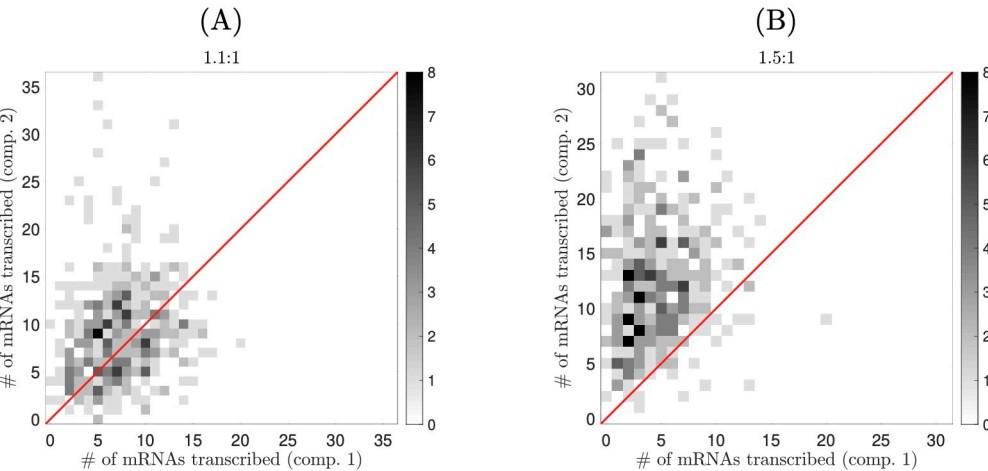

**Fig 8. Compartment size differences skew the relative number of mRNAs transcribed by each nucleus.** Number of mRNAs transcribed in each nuclear compartment for approximately 400 circadian periods in two different two-nucleus syncytia (panel titles indicate size ratios). The red line represents equal numbers of mRNAs being produced by each nucleus.

To further study division of labor for unequal compartment sizes, we repeated the simulation from the previous section, this time for a variety of two-compartment model syncytia, in which the larger compartment was 1.1, 1.2, . . ., 1.5 times the length of the smaller compartment. As before, simulations were run for 10000 $h$, with transcription rate $\alpha = 18\ h^{-1}$. In Fig 8, we display scatter plots for the 1.1x and 1.5x cases. We focus specifically on the skew in the number of transcribed mRNAs—i.e., the fraction of mRNAs that are transcribed by the nucleus in the smaller compartment. Although we find that mRNA product is unevenly distributed between nuclei from cycle to cycle, uneven compartment sizes consistently bias mRNA production from cycle to cycle. Mean mRNA production skews slightly towards the smaller compartment in the 1.1x case, and dramatically towards the smaller compartment in the 1.5x case (data on mean transcription fractions is shown for all cases in Table 3).

Is limit cycle quality substantially reduced when mRNA production asymmetries are induced by changing compartment sizes? To answer this question, we ran 100 realizations of our stochastic model for each of the two-compartment model syncytia outlined above; each realization was run for 1000 $h$. Since quality factor varies with compartment size, we compute the quality factor for the entire cell in each case (i.e. we find the quality factor for the *total* number of nuclear proteins) rather than for a single compartment. We then compare this set of quality factors to the factor for an entire cell composed of two compartments of equal length. We find that increasing variability in compartment length has a very minimal effect on

**Table 3. Ratio of compartment lengths $l_1/l_2$, along with fraction of total cytoplasmic volume occupied by compartment 2, vs. fraction of total mRNAs transcribed in compartment 2 (the smaller compartment).**

| $l_1/l_2$ | $l_2/(l_1 + l_2)$ | mean fraction of mRNAs transcribed in comp. 2 |
|---|---|---|
| 1.1 | 0.476 | 0.559 |
| 1.2 | 0.455 | 0.600 |
| 1.3 | 0.435 | 0.654 |
| 1.4 | 0.417 | 0.688 |
| 1.5 | 0.400 | 0.731 |

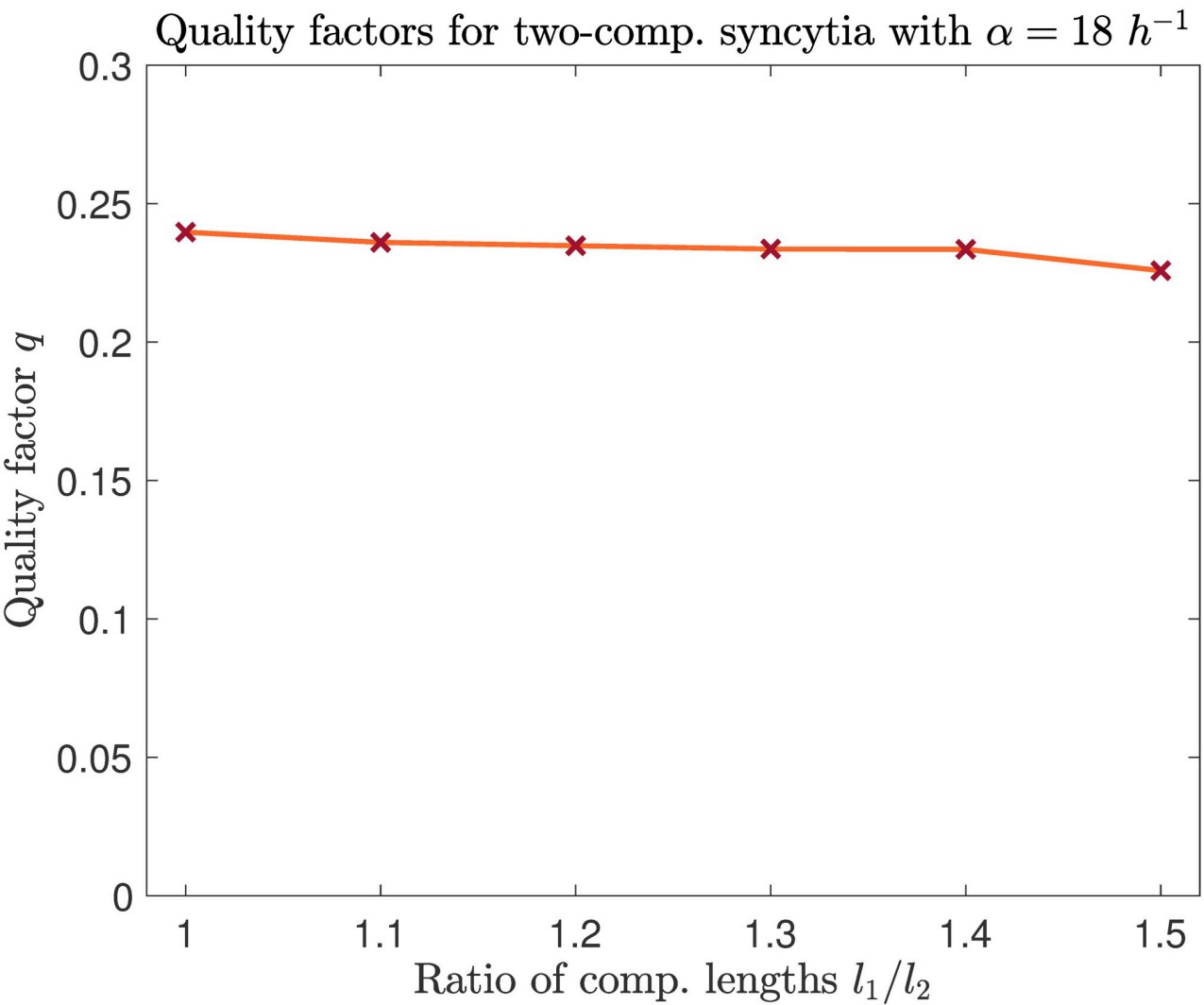

**Fig 9. Quality factor (across the entire cell) is close to uniform for a variety of two-nucleus syncytia.** We varied the asymmetry of compartment sizes and measured the mean quality of the oscillator over 1000 simulated hours, with $\alpha = 18\ h^{-1}$. Quality factor varies little and consistently exceeds the value for a uninucleate cell (0.192).

limit cycle quality (Fig 9). Our results show that even with extreme division of labor between nuclei, a high quality oscillator can be assembled. The nucleus in the larger compartment, though it produces the minority of the cell's mRNAs, contributes enough to ensure that the quality factor remains consistently larger than the factor for a uninucleate cell (0.192 for the parameter values assayed here).

**Limit cycle consistency in a model syncytium.** Lastly, we wish to examine whether protein sharing supports consistency of circadian timekeeping in a model syncytium when transcription rate is very low ($\alpha = 18\ h^{-1}$). We return to uniform compartment sizes and consider two different scenarios: (i) eight nuclear compartments that share proteins via diffusion and (ii) eight independent nuclear compartments, with each compartment containing only the mRNAs and proteins produced by its nucleus. In both scenarios, every compartment is given

the same initial mRNA and protein counts, i.e.,

$$m_n^{(1)}(0) = m_n^{(2)}(0) = \ldots = m_n^{(8)}(0) = \mu_n, \quad m_c^{(1)}(0) = m_c^{(2)}(0) = \ldots = m_c^{(8)}(0) = \mu_c,$$

$$p_c^{(1)}(0) = p_c^{(2)}(0) = \ldots = p_c^{(8)}(0) = \rho_c, \qquad p_n^{(1)}(0) = p_n^{(2)}(0) = \ldots = p_n^{(8)}(0) = \rho_n,$$

where $\mu_n$, $\mu_c$, $\rho_c$, and $\rho_n$ are integer constants. Starting from these initial conditions, we simulate five circadian "days" for the two different scenarios outlined above. In this context, a "day" refers to the time period between peaks in total nuclear protein expression (i.e., the total nuclear protein count for the entire cell). We are interested in evaluating the consistency in the timing of these peaks. We therefore ran 100 stochastic simulations for each scenario, with the same initial conditions for every trial, and recorded the time that each peak occurred for each simulation.

In Fig 10, we display histograms for the times that peaks occurred, where time 0 $h$ indicates the time for the first peak. Unsurprisingly, we find that the timing of peaks becomes more unpredictable over time. However, since the nuclei from scenario (i) share proteins, oscillations for nuclear protein in each compartment remain roughly in-phase from their initially synchronized states. On the other hand, in scenario (ii), nuclei have no means to "communicate," so, even though they begin synchronized, nuclear protein oscillations drift quickly out of phase. Hence, timing of peaks in *total* nuclear protein is considerably more unpredictable in scenario (ii) than in scenario (i), as indicated by larger standard deviations for peak times in scenario (ii) (Fig 10). This shows that protein diffusion can help regulate circadian timekeeping in a syncytial cell, and that circadian rhythms quickly break down in a syncytial cell with no communication between its nuclei. In the absence of external cues such as light, circadian rhythms tend to deteriorate after several days [16, 31]. However, we've shown that, like light, protein sharing between nuclei can have an entraining (synchronizing) effect on the cell's rhythms, helping maintain reliable rhythms for a longer period of time.

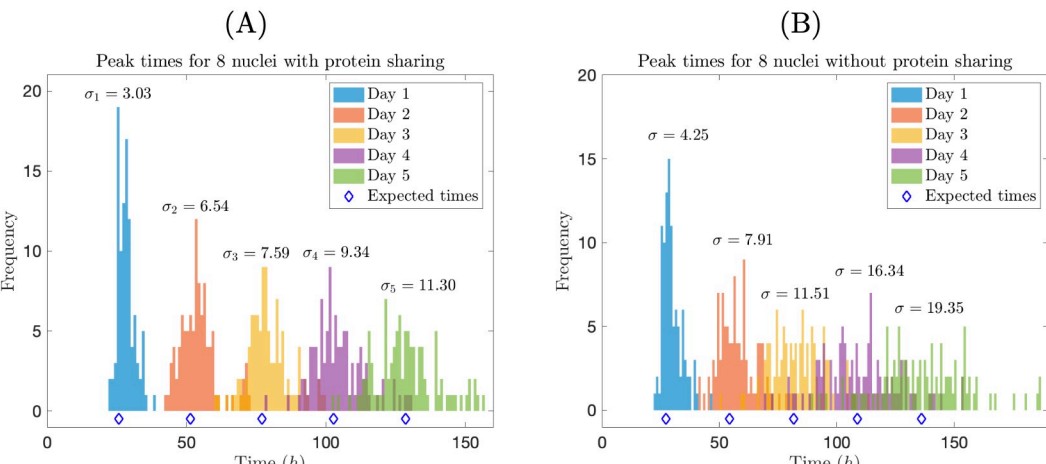

**Fig 10. Protein sharing supports a consistent circadian rhythm.** Times of peaks of total nuclear protein expression, for eight nuclear compartments that share proteins (A) and eight independent nuclear compartments (B). We ran 100 simulations in each case, with $\alpha = 18 \ h^{-1}$. In each figure, we also indicate the standard deviation $\sigma_i$ of peak time for each circadian "day" ($i = 1, 2, \ldots, 5$). Expected times for peaks (i.e., average circadian day lengths) are indicated with blue diamonds.

## Discussion

We have formulated and simulated a mathematical model for circadian rhythms in a syncytial cell, adapted from the uninucleate model of Wang and Peskin [21]. The motivation for developing our model was to better understand division of labor (i.e., partitioning of mRNA transcription) between nuclei in *Neurospora crassa*. Our results indicate that, by "sharing" proteins between nuclei via molecular diffusion, nuclei within a syncytial cell such as *Neurospora* can achieve a robust, reliable circadian rhythm with minimal mRNA production. For example, we found that a model syncytium with eight nuclei can achieve nuclear protein oscillations of comparable "quality" to a uninucleate cell while reducing transcription rates by a factor of one thousand. This is because protein sharing has an averaging effect on the model, removing much of the consequences of random mRNA fluctuations. While our model is undoubtedly simplified, it offers a potential explanation of recent experimental results, which suggest that *Neurospora* achieves a robust circadian rhythm with very small mRNA counts (Brad Bartholomai, personal communication, 2019).

Many of the simulations in this paper were run below the deterministic threshold for a Hopf bifurcation derived in [21] (see the inequality (6)). We selected this regime of the model because we wanted to test whether oscillations could be achieved at the very low transcription rates that are consistent with smFISH measurements of mRNA in *Neurospora crassa* (Brad Bartholomai, personal communication, 2019). Remarkably, we found that, even in uninucleate systems, quasi-periodic limit cycles are attained in stochastic simulations when the transcription rate is two orders of magnitude lower than this threshold and mRNA counts are in the single digits. However, absolute quantification of proteins is very difficult in syncytial cells [32], so the actual translation rates in *Neurospora* may be considerably higher than the parameter value we used. A high translation rate $\beta$ could potentially push the system back into the regime where a deterministic limit cycle exists. Regardless, it is notable that random transcription events can potentially maintain limit cycles for very low transcription rates, allowing a cell to achieve a robust circadian rhythm with minimal labor upon its nuclei.

Extreme mRNA efficiency may confer fitness advantages upon fungal cells. These cells are often regarded as enzyme factories—capable of expressing vast quantities of potentially useful proteins [33]. Although the energetic cost of mRNA transcription is not large, the physical rearrangements needed to access a particular gene may create interference when a nucleus must access multiple different regions of its genome to express multiple genes [34]. Reducing transcription rates on each gene may therefore allow a nucleus to transcribe a larger set of genes. In Zaslaver et al. [34], the authors highlight that organisms may benefit from minimizing the rates of transcription, while keeping protein abundances constant. The super-secretory abilities and rapid growth of fungi may emerge from their ability to push transcription rates to extremely low rates while using protein sharing to suppress stochastic fluctuations in the abundances of the proteins produced.

Many previous mathematical models have addressed the synchronization of coupled circadian oscillators [26, 35–38]. These studies have shown that intercellular coupling leads to a more consistent, noise-resistant, and robust circadian rhythm [26, 37, 38] and that coupled circadian oscillators are capable of entrainment to light-dark cycles [35–37]. Liu et al. [39] used bioluminescence imaging to show that intercellular coupling in the suprachiasmatic nucleus is essential to synchronize cellular oscillators and provide robustness against genetic mutations. Our model is notable because it demonstrates that coupling of nuclei (via protein sharing) *within* a cell can provide similar benefits: namely, it can increase the consistency of period length and the robustness of oscillations, while reducing the detrimental effects of noise.

In constructing our syncytial cell model, we made the simplifying assumption that protein diffusion is "fast" relative to protein import and decay. Hence, we imposed that protein levels are constantly uniformized between compartments. While this assumption may be fairly reasonable across a small number of nuclear compartments, it breaks down for a larger cell with many nuclear compartments. So, while our model predicts that limit cycle quality increases with number of nuclei (Fig 4), is there a trade-off that occurs in a larger cell, as communication between nuclei becomes more limited? A model that includes more accurate diffusive mechanics could help answer this question, and better predict how spacing of nuclei across the cell affects the quality of circadian rhythms.

In future, it may be valuable to adapt our syncytial model to more closely describe circadian rhythms in *Neurospora*. *frq* mRNA and protein oscillations in *Neurospora* are driven by interlocking positive and negative feedback loops, while our model involves only a negative feedback loop. A minimal differential equation model to study *Neurospora* circadian rhythm would likely need to include *frq* mRNA and protein, White Collar mRNA and protein, the protein interactions to produce the FRQ-White Collar complex (whose formation inhibits *frq* transcription), and measured values for the transcription and export rates of mRNAs to quantitatively match to the emerging data streams on real mRNA and protein abundances [40, 41]. At the same time, *Neurospora's* ability to generate heterokaryotic syncytia—that is, syncytia from genetically divergent nuclei—allows for thorough probing of the roles of nuclear interactions in circadian timing, by comparing syncytial performance when all nuclei participate in *frq* transcription with syncytia in which only some fraction of nuclei can transcribe the requisite mRNAs.

## Supporting information

**S1 Appendix. Fast protein diffusion justifies treating compartmental protein concentrations as uniform.**
(PDF)

## Author Contributions

**Conceptualization:** Leif Zinn-Brooks, Marcus L. Roper.

**Data curation:** Leif Zinn-Brooks, Marcus L. Roper.

**Formal analysis:** Leif Zinn-Brooks, Marcus L. Roper.

**Funding acquisition:** Marcus L. Roper.

**Investigation:** Leif Zinn-Brooks, Marcus L. Roper.

**Methodology:** Leif Zinn-Brooks, Marcus L. Roper.

**Visualization:** Leif Zinn-Brooks, Marcus L. Roper.

**Writing – original draft:** Leif Zinn-Brooks, Marcus L. Roper.

**Writing – review & editing:** Leif Zinn-Brooks, Marcus L. Roper.

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
