## [Decision Letter · Decision Letter 0]

12 Mar 2021

Dear Dr. Roper,

Thank you very much for submitting your manuscript "Circadian rhythm shows potential for mRNA efficiency and self-organized division of labor in multinucleate cells" for consideration at PLOS Computational Biology.

As with all papers reviewed by the journal, your manuscript was reviewed by members of the editorial board and by several independent reviewers. In light of the reviews (below this email), we would require a significantly-revised version that takes into account the reviewers' comments.

If you choose to revise and resubmit the manuscript, your revised manuscript will be sent to reviewers for further evaluation, so please address reviewer comments thoroughly.

Sincerely,

William Cannon

Guest Editor

PLOS Computational Biology

Mark Alber

Deputy Editor

PLOS Computational Biology

Reviewer's Responses to Questions

**Comments to the Authors:**

Reviewer #1: Attached

Reviewer #2: In this manuscript the authors describe a mathematical model on the syncytical organization of circadian rhythms by using Neurospora crassa as an example. This is an interesting and important aspect about the circadian organization in Neurospora and fungi.

However, when assessing their model, there appear to be several shortcomings and perhaps misconceptions.

First of all, I would have appreciated to see a scheme, i.e. a picture representation of the compartments together with the molecular processes inside those and between them, which define the model.

The model itself seems to be quite restrictive. While proteins are allowed to diffuse between different compartments, Eq 3 does not seem to include the situation that a protein from compartment i, but now due to rapid diffusion being in compartment j can diffuse into the nucleus of compartment j. I think, when allowing (rapid) diffusion between compartments, Eq 3 needs to be changed such that import of proteins from neighboring compartments can be included.

The other problem I have with the model is that mRNA's are not allowed to diffuse into other compartments. Why is that? From a molecular mass perspective clock-mRNA (frq-mRNA) may diffuse at least as fast as proteins. Since the manuscript concludes with an "extreme RNA efficiency", this mRNA-efficacy may occur as a result to the restrictions imposed on the model.

Next (below) Eq 3 the authors write:

"In contrast to nucelar mRNA, however, cytoplasmic proteins can also move between compartments via diffusion".

Please note that the mRNA is exported from the nucleus into the cytosol where protein synthesis of proteins (for example FRQ) occurs. It should also be noted that the FRQ protein in Neurospora crassa is present as a dimer. This dimer formation may have some influence when considering heterokaryons.

An interesting aspect of the syncytium in a circadian context is when heterokaryons are considered, i.e. when the nuclei contain different alleles, for example from a long period mutant (frq7) and a short period mutant (frq1) of Neurospora crassa. Loros et al. (Genetics 114:1095ff, 1986) studied the circadian behavior of such heterokaryons, which in case of frq1 and frq7 showed an interesting superposition of the respective period lengths for different frq1 and frq7 heterkaryon mixes. I feel that such data could be used to "benchmark" a syncytical model with respect to circadian rhytmicity.

Table 1: The change of alpha by four-orders of magnitude should have a significant effect on the period length of the oscillator when only alpha is changed. What is the effect of alpha on the period?. While one can agree on the change on q, what does such a calculation mean? Are in the model other parameters changed to keep the period within the circadian range when alpha is increased?

The model as such, although the calculations appear expertly done, suffers from a clear correlation between model assumptions made, computational results and corresponding experimental facts/knowledge that may support the model implications. My overall feeling is that the model is premature.

Minor things:

Abstract, line 4: nuclei are capable of autonomous behavior. What kind of autonomous behavior?

Nuclei in some studied fungi appear to divide in synchrony.

Introduction, line 47 and several other places in the ms: why should Poisson noise be considered? Are there other types of noise that need to be excluded?

**Have all data underlying the figures and results presented in the manuscript been provided?**

Reviewer #1: Yes

Reviewer #2: Yes

PLOS authors have the option to publish the peer review history of their article (what does this mean?). If published, this will include your full peer review and any attached files.

Reviewer #1: No

Reviewer #2: No
---

## [Decision Letter · Decision Letter 1]

12 Jul 2021

Dear Dr. Roper,

We are pleased to inform you that your manuscript 'Circadian rhythm shows potential for mRNA efficiency and self-organized division of labor in multinucleate cells' has been provisionally accepted for publication in PLOS Computational Biology.

Best regards,

William Cannon

Guest Editor

PLOS Computational Biology

Mark Alber

Deputy Editor

PLOS Computational Biology

Reviewer's Responses to Questions

**Comments to the Authors:**

Reviewer #1: The authors fully respond to all my comments.

Reviewer #2: The authors have answered my queries in a satisfactory way. Although one may pick on certain experimental aspects omitted, for example with respect to protein (FRQ)-dimer formation, I feel the manuscript makes an interesting attempt to understand syncytial organization.

**Have the authors made all data and (if applicable) computational code underlying the findings in their manuscript fully available?**

Reviewer #1: None

Reviewer #2: Yes

PLOS authors have the option to publish the peer review history of their article (what does this mean?). If published, this will include your full peer review and any attached files.

Reviewer #1: No

Reviewer #2: **Yes: **Peter Ruoff

---

## [Editor Report · Acceptance letter]

28 Jul 2021

PCOMPBIOL-D-21-00232R1 

Circadian rhythm shows potential for mRNA efficiency and self-organized division of labor in multinucleate cells

Dear Dr Roper,

I am pleased to inform you that your manuscript has been formally accepted for publication in PLOS Computational Biology. Your manuscript is now with our production department and you will be notified of the publication date in due course.

With kind regards,

Andrea Szabo
